# Flower Extracts from Ornamental Plants as Sources of Sunscreen Ingredients: Determination by In Vitro Methods of Photoprotective Efficacy, Antigenotoxicity and Safety

**DOI:** 10.3390/molecules27175525

**Published:** 2022-08-27

**Authors:** Jorge Luis Fuentes, Carlos Adolfo Pedraza Barrera, Diego Armando Villamizar Mantilla, Silvia Juliana Flórez González, Lady Johanna Sierra, Raquel Elvira Ocazionez, Elena E. Stashenko

**Affiliations:** 1Grupo de Investigación en Microbiología y Genética (GIMG), Universidad Industrial de Santander, Bucaramanga 680002, Colombia; 2Centro de Investigación en Biomoléculas (CIBIMOL), Universidad Industrial de Santander, Bucaramanga 680002, Colombia; 3Centro de Cromatografía y Espectrometría de Masas (CROM-MASS), Universidad Industrial de Santander, Bucaramanga 68000, Colombia

**Keywords:** ultraviolet light, photoprotection, antigenotoxicity, cytotoxicity, genotoxicity, human fibroblasts, ornamental plants

## Abstract

Plants are sources of sunscreen ingredients that prevent cellular mutations involved in skin cancer and aging. This study investigated the sunscreen properties of the extracts from some ornamental plants growing in Colombia. The UV filter capability of the flower extracts obtained from *Rosa centifolia* L., *Posoqueria latifolia* (Rudge) Schult, and *Ipomoea horsfalliae* Hook. was examined. Photoprotection efficacies were evaluated using in vitro indices such as sun protection factor and critical wavelength. UVB antigenotoxicity estimates measured with the SOS Chromotest were also obtained. Extract cytotoxicity and genotoxicity were studied in human fibroblasts using the trypan blue exclusion and Comet assays, respectively. Major compounds of the promising flower extracts were identified by UHPLC–ESI+–Orbitrap–MS. The studied extracts showed high photoprotection efficacy and antigenotoxicity against UVB radiation, but only the *P. latifolia* extract showed broad-spectrum photoprotection at non-cytotoxic concentrations. The *P. latifolia* extract appeared to be safer for human fibroblast cells and the *R. centifolia* extract was shown to be moderately cytotoxic and genotoxic at the highest assayed concentrations. The *I. horsfalliae* extract was unequivocally cytotoxic and genotoxic. The major constituents of the promising extracts were as follows: chlorogenic acid, ecdysterone 20E, rhamnetin-rutinoside, *cis*-resveratrol-diglucoside, *trans*-resveratrol-diglucoside in *P. latifolia*; quercetin, quercetin-glucoside, quercetin-3-rhamnoside, kaempferol, kaempferol-3-glucoside, and kaempferol-rhamnoside in *R. centifolia*. The potential of the ornamental plants as sources of sunscreen ingredients was discussed.

## 1. Introduction

Photoprotection is a preventive strategy to defend human skin against cancer and photoaging [1]. The sunlight UV rays that reach the Earth’s surface, such as ultraviolet A (320–400 nm) and ultraviolet B (280–320 nm), cause DNA damage (e.g., cyclobutane pyrimidine dimers), which initiates inflammatory processes and skin cancer [2,3]. The use of sunscreens is among the most popular strategies in photoprotection [4]. Sunscreens contain compounds that act like filters, either absorbing or refracting the UV radiation [5]. Relevant properties of sunscreens used commercially have been reviewed [6,7].

It has been indicated that active ingredients of commercial sunscreens can be toxic to humans and coral reefs [8,9,10,11], which has increased the interest in using phytochemicals in sunscreen formulations [12,13,14,15,16,17]. Phytochemicals are genoprotective and anticancer agents [18,19] that are considered to be non-toxic and pharmacologically safe for human consumption [13]. These phytochemicals can prevent cellular mutations involved in skin cancer and aging by regulating UV-induced mutability [20,21].

Among the wide plant diversity, ornamental species have been widely used as raw materials for the cosmetic industry due to both their fragrance and therapeutic properties [22]. Colombia is the second highest exporter of ornamental flowers worldwide [23]; however, until today, their use has been exclusively for decorative purposes. In the present study, we took advantage of the cosmetic and therapeutic properties of ornamental plants to find phytochemicals that can be used as sunscreen ingredients.

The work aims were as follows: (i) to evaluate the UV absorption capability of flower extracts obtained from some ornamental plants growing in Colombia by means of in vitro protection efficacies (SPF_in vitro_, and λc), (ii) to study the correlation between in vitro UVB protection efficacy (SPF_in vitro_) and complementary SOS Chromotest-based DNA protection using data from flower extracts, (iii) to evaluate in human fibroblast the cytotoxicity and genotoxicity of promising extracts, and (iv) to study the chemical composition of promising extracts as sources of sunscreens. We showed that the flower extracts isolated from ornamental plants, *R. centifolia* and *P. latifolia*, were rich in photoprotective compounds. *P. latifolia* extract, which is photostable and relatively safe, seems to be a good candidate for a sunscreen active ingredient.

## 2. Results

### 2.1. In Vitro Photoprotection Efficacy of the Flower Extracts

The UV absorbance spectrum of each flower extract is shown in Figure 1; high levels of absorbance across the UV spectrum were observed for all extracts. The extracts showed the highest absorbance peaks for λ between 225 and 320 nm. All the absorbance peaks depended on the extract concentrations.

The SPF_in vitro_ values calculated for each flower extract are presented in Table 1. According to the European Commission categories (see Section 4), the four flower extracts showed UVB photoprotection efficacy (SPF_in vitro_ ≥ 6.0); the SPF_in vitro_ values increased with concentration (R = 0.97, *p* ˂ 0.001, Figure 2A). Except for the *I. horsfalliae* extract, other extracts also showed broad-spectrum (UVA-UVB) protection efficacy (λc ≥ 370 nm) at a high extract concentration (750 µg/mL).

### 2.2. Relations between SPF_in vitro_ and %GI Estimates in Flower Extracts

None of the flower extracts had increased *I* values at concentrations assayed in the SOS Chromotest (data not shown); therefore, they were considered not genotoxic in the model *E. coli* PQ37 cells. All photoprotective extracts also showed antigenotoxic against UVB radiation (Table 1). The SPF_in vitro_ and %GI values in flower extracts were highly correlated (R = 0.82, *p* ˂ 0.001, Figure 2B). That is, the greater the UVB photoprotective efficiency, the lower the genetic damage.

### 2.3. Extract Cytotoxicity in Human Fibroblast (MRC5) Cells

Cytotoxicity dose–response relations were studied for each flower extract at a concentration range between 62 and 750 µg/mL (Table 1). Next, lethal concentrations, 50% (LC_50_) and 30% (LC_30_), in human fibroblast (MRC5) cells were obtained by interpolation. Based on the LC_50_ values, extract cytotoxicity was as follows: *I. horsfalliae* (398 µg/mL) ˃ *R. centifolia* pink, commercial variety (492 µg/mL) ˃ *R. centifolia* fuchsia, commercial variety (702 µg/mL). The extracts were safe for fibroblast cells at concentrations lower than LC_30_ values as follows: *I. horsfalliae* (250 µg/mL); *R. centifolia* pink, commercial variety (363 µg/mL); and *R. centifolia* fuchsia, commercial variety (492 µg/mL). The *P. latifolia* extract was unique in that it was shown to be non-cytotoxic to fibroblast cells at the concentration range studied. All the extracts were relatively less cytotoxic than the commercial sunscreen (Eau Thermale Avène SPF 50+) and the sunscreen active ingredient (titanium dioxide) used for comparison.

### 2.4. Extract Genotoxicity in Human Fibroblast (MRC5) Cells

Genotoxicity dose–response relations were studied for each flower extract (Table 2). According to genotoxicity criteria, the extracts produced some degree of DNA damage at the following concentrations: *I. horsfalliae* (187.5 µg/mL) ˃ *R. centifolia* (375.0 µg/mL) ˃ *P. latifolia* (750 µg/mL). Evaluation of the equivalent solvent concentrations (dilutions) in extracts indicated that the solvent (methanol) was not genotoxic in human fibroblasts for the concentration range tested. Therefore, except for *P. latifolia*, the extracts showed low-to-moderate genotoxicity across the concentration range studied. Such genotoxicity showed a clear dose–response relationship (extract concentration–DNA damage), and demonstrates the importance of establishing the safe extract concentrations for potential use as sunscreen ingredients. These data also suggest a cytotoxic mode of action depending on the genotoxicity for all extracts.

### 2.5. In Vitro Photoprotection Efficacy and Photostability at Safe Extract Concentrations

At photoprotective and safe extract concentrations, namely, at non-cytotoxic (≤LC_30_) and non-genotoxic concentrations, only the *P. latifolia* extract showed high UVB and broad-spectrum (UVA-UVB) photoprotection efficacy values (Table 1). The other extracts demonstrated a reduction in their SPF_in vitro_ values, which indicates their low photoprotection efficacy (6.0 ≤ SPF_in vitro_ ≤ 15.0; λc ˂ 370 nm) values. Conversely, extract photostability or effectiveness (E_ff_) when they were irradiated at Fitzpatrick’s MDE was consistently high at photoprotective and safe extract concentrations. According to these results, the *P. latifolia* and *R. centifolia* extracts at non-cytotoxic concentrations could be good candidates for use as sunscreen active ingredients. Conversely, the *I. horsfalliae* extract at non-cytotoxic concentrations was shown to be genotoxic (Table 2) and poorly photoprotective (Table 1); therefore, this extract was excluded from further analyses.

### 2.6. Chemical Characterization of the Promising Flower Extracts by UHPLC–ESI+–Orbitrap–MS

The yields of the hydroalcoholic flower extracts were as follows: *P. latifolia* (41.2 ± 0.1%) ˃ *R. centifolia* fuchsia, commercial variety (30.9 ± 0.0%) ˃ *R. centifolia* pink, commercial variety (18.2 ± 0.1%). The UHPLC–ESI+–Orbitrap–MS analysis of the flower extracts from *R. centifolia* and *P. latifolia* plant species allowed us to identify presumptively several compounds on the basis of their mass spectra fragmentation patterns and exact mass measurements (Table 3).

The major compounds in the extracts (≥17 mg ± SD/g of extracts) were as follows: quercetin-3-rhamnoside (49 ± 2), kaempferol-3-glucoside (70 ± 12), kaempferol-rhamnoside (64 ± 8), quercetin (160 ± 26), and kaempferol (146 ± 5) in *R. centifolia* pink, commercial variety; cyanidin-3,5-glucoside (34 ± 1), quercetin-glucoside (17 ± 1), quercetin-3-rhamnoside (32 ± 1), kaempferol-3-glucoside (41 ± 1), kaempferol-rhamnoside (23 ± 2), quercetin (130 ± 14), and kaempferol (51 ± 4) in *R. centifolia* fuchsia, commercial variety; chlorogenic acid (35 ± 1), ecdysterone (64 ± 8), rhamnetin-rutinoside (17 ± 1), *cis*-resveratrol-diglucoside (140 ± 7), and *trans*-resveratrol-diglucoside (280 ± 16) in *P. latifolia*. Other compounds were also detected in the extracts at a minor proportion (˂17 mg ± SD/g of extracts), as follows: quercetin-rutinoside-rhamnoside (1.5 ± 0.1), kaempferol-rhamninoside (3.7 ± 0.4), ramnetin-rhamnoside (1.7 ± 0.1), quercetin-3-rutinoside (6.9 ± 0.4), kaempferol-neohesperidoside (8.7 ± 0.5), and quercetin-arabinoside (1.4 ± 0.2). The chromatographic profiles obtained for the flower extracts studied, using the UHPLC–ESI+–Orbitrap–MS technique, are shown in Figure 3.

## 3. Discussion

The present work evidenced that the flower extracts isolated from several ornamental plants cultivated in Colombia contained compounds that could be used as sunscreen ingredients. Among the eleven plant species studied in our project (data not shown), three plants (*R. centifolia*, *P. latifolia*, and *I. horsfalliae*) in particular showed good photoprotective properties. This finding provides new evidence on plant extract applicability as a source of solar filters (Table 4).

We also showed that the photoprotection efficacy depended on extract concentration. The photoprotective extracts were also antigenotoxic against UVB. This indicates that they acted as filters that absorbed or refracted the UV rays and reduced genotoxicity. As in the previous studies [17,31,32,33], our data support the use of DNA damage detection assay (in this case, the SOS Chromotest) as an effective complement that improves the efficacy of photoprotection measurement.

**Table 4 molecules-27-05525-t004:** Some plant species with reported sunscreen properties ^†^.

Plant Family	Species Name	UV Protective Rank	References
Adoxaceae	*Sambucus nigra*	UVA	Jarzycka et al. [34]
Asteraceae	*Achyrocline satureioides*	UVB	Fuentes et al. [17]
	*Baccharis antioquensis*	UVA-UVB	Mejía-Giraldo et al. [14,35,36]
	*Chromolaena pellia*	UVA-UVB	Fuentes et al. [17]
	*Helichrysum arenarium*	UVA	Jarzycka et al. [34]
	*Pentacalia pulchella*	UVA-UVB	Mejía-Giraldo et al. [14,35,37]
Bromeliaceae	*Neoglaziovia variegata*	UVB	de Oliveira-Junior et al. [38]
Calophyllaceae	*Calophyllum inophyllum*	UVA-UVB	Ku et al. [39]
Convolvulaceae	*Ipomoea horsfalliae*	UVB	Sierra et al. [40]
Cucurbitaceae	*Momordica charantia*	UVB	Guimarães de Sousa et al. [41]
Clusiaceae	*Garcinia brasiliensis*	UVB	Figueiredo et al. [42]
Fabaceae	*Bauhinia microstachya*	UVB	Reis-Mansur et al. [43]
	*Dimorphandra gardneriana*	UVB	Nunes et al. [15]
Myricaceae	*Morella parvifolia*	UVA-UVB	Puertas-Mejía et al. [44]
Nycataginaceae	*Boerhavia diffusa*	UVB	Guimarães de Sousa et al. [41]
Rosaceae	*Crataegus monogyna*	UVA	Jarzycka et al. [34]
Verbenaceae	*Lippia microphylla*	UVB	Nunes et al. [15]
	*Lippia origanoides*	UVB	Fuentes et al. [17]
Vitaceae	*Vitis vinifera*	UVA-UVB	Hübner et al. [45]

^†^, Modified from Fuentes et al. [17].

As we indicated in the Introduction, some organic ingredients of commercial sunscreens can be toxic to humans and coral reefs [8,9,10], with inorganic filters (i.e., titanium dioxide or zinc oxide) being a safer alternative [11]. At photoprotective concentrations, the extracts were relatively less cytotoxic than a commercial sunscreen (Eau Thermale Avène SPF 50+), and its active ingredient (titanium dioxide) was used for comparison. In this sense, the plants studied here could be new and safer sources of ingredients for the development of commercial sunscreens.

The *P. latifolia* extract was the most promising for use as a sunscreen, because it showed high photoprotective efficacy, antigenotoxicity, photostability, and relatively low cytotoxicity and genotoxicity in human fibroblasts. *P. latifolia*, commonly known as jasmine tree—mountain lily (*azuceno*)—monkey apple, is a native plant from Central and South America [46]. This tree possesses flowers with a very intense scent, whose main fragrance compound is (*Z*)-3-hexenyl acetate [47]. We found that *P. latifolia* ethanolic extract was rich in *cis*- and *trans*-resveratrol diglucosides, two stilbenoid isomers, which were previously identified in *Vitis vinifera* [48] and *Glycine max* [49]. Chlorogenic acid (35 ± 1 mg/g), ecdysterone (64 ± 8 mg/g), and rhamnetin-rutinoside (17 ± 1 mg/g) were present as well at relevant concentrations. Chlorogenic acid, resveratrol, and its derivatives have been reported as compounds with solar filter activity [50,51,52,53]. Resveratrol and its derivatives have also shown antigenotoxicity against UV radiation and skin cancer chemopreventive properties [54,55,56]. We hypothesized that some of these compounds, or their combinations, were responsible for the sunscreen/antigenotoxic properties of the extracts studied here. Among the plant species studied, the *P. latifolia* extract appeared to be the best candidate and the safest for use as a sunscreen active ingredient.

The flower extracts from *R. centifolia* had high UVB and broad-spectrum protection efficacies as well. However, these extracts showed between low and moderate cytotoxicity and genotoxicity in human fibroblasts measured at the highest concentrations compared with those *Rosa* species extracts previously studied [57]. This supported the importance of establishing safe extract concentrations for use of the extracts as sunscreen ingredients, as has been previously suggested [22]. Thus, these extracts should be used with caution until more details on their genotoxicity are known. The *R. centifolia* extracts were rich in kaempferol and quercetin and their derivatives (quercetin-glucoside, quercetin-3-rhamnoside, kaempferol-3-glucoside, kaempferol-rhamnoside), as has also been found for other *Rosa* species [26,58,59,60,61]. For *R. centifolia* species, the antimutagenic activity has been previously reported against ethyl methanesulfonate [62]. Kaempferol and quercetin showed solar filter activity [52,53,63]. These main compounds were possibly responsible for the sunscreen/antigenotoxic properties of the *R. centifolia* extracts studied in this work.

Conversely, the extract obtained from *I. horsfalliae,* a species with previously reported sunscreen properties [40], was shown to be cytotoxic and genotoxic to human fibroblast cells. Little is still known about the genotoxicity of the major compounds (chlorogenic acid, dicaffeoylquinic acid, and scopoletin) of this extract. Chlorogenic acid and scopoletin have shown clastogenic activity [64,65], which suggests that flower extract’s genotoxic activity could be related to these compounds, although, a synergistic effect of these compounds could also be the cause of their genotoxicity. Confirmation of this hypothesis requires further studies.

## 4. Materials and Methods

### 4.1. Plant Material and Extracts

The flowers from *Rosa centifolia* (Rosaceae), pink and fuchsia commercial varieties, were supplied by Flexport—Colombia S.A.S. (Bogotá, Cundinamarca, Colombia). The flowers from *Posoqueria latifolia* (Rubiaceae) and *Ipomoea horsfalliae* (Convolvulaceae) species were collected from experimental plots at the Agroindustrial Pilot Complex of the National Center for Agroindustrialization of Aromatic and Medicinal Tropical Vegetal Species (CENIVAM). The *P. latifolia* e *I. horsfalliae* species were identified at the Colombian National Herbarium, where their vouchers (COL; voucher number in parentheses) were placed (Table 1).

For each specimen, undamaged and fully developed flowers were dried in an Advantage Plus Tray Lyophilizer (Virtis Co., Gardiner, ME, USA) and were used for solvent extraction as indicated by Sierra et al. [40]. In brief, dried flowers (1 g) were mixed with acidified ethanol solution (20 mL, 0.5% HCl, 1:1 *v*/*v*) and put for 5 min in an S15H ultrasound bath (Elmasonic, Singen, Germany). The mixture was filtered, and the residue was extracted twice more. Extracts were roto-evaporated and then were dried as indicated above. Extract stock solutions were prepared from the dried powder (30 mg), which were dissolved in methanol (1 mL), vortexed (3 min), exposed to ultrasound (10 min, 40 °C), and centrifuged (5000× *g*, 10 min). The supernatant (1 mL) was then filtered and was stored at −80 °C in a Thermo Scientific^®^ Series-86 DEG C ultra-low-temperature freezer (Thermo Scientific, Waltham, MA, USA). Before their use, the extract stock solutions were defrosted and refrigerated (5–8 °C) for 24 h.

### 4.2. Chemicals, Buffer, Enzymes, and Culture Media

The titanium dioxide, Luria–Bertani (LB) media, antibiotics (ampicillin and tetracycline), trypan blue solution (0.4%), Bioultra lyophilized proteinase K, and high-resolution agarose were obtained from Sigma-Aldrich Corp. (Milwaukee, WI, USA). The YOYO solution was purchased from Thermo Scientific (Waltham, MA, USA). The substrates for β-galactosidase (*o*-nitrophenyl-β-d-galactopyranoside) and alkaline phosphatase (*p*-nitrophenylphosphate) were purchased from Amresco (Slon, OH, USA). The Dulbecco’s modified Eagle medium (DMEM), Ham’s nutrient mixture F12 (F-12), fetal bovine serum (FBS), phosphate-buffered saline (PBS), trypsin EDTA solution, and antibiotics (penicillin–streptomycin mixture) were acquired from Gibco (Grand Island, NY, USA). Other reagents and solvents were obtained from commercial houses J.T. Baker (Phillipsburg, NJ, USA) or Merck (Kenilworth, NJ, USA).

### 4.3. UV Absorption Capability

Aqueous aliquots (1.5 mL) of each extract, prepared at different concentrations (between 62.5 and 750.0 μg/mL) using distilled water, were placed in a quartz cuvette (1 cm step length and 1.5 mm glass thickness), and their UV absorption spectra (λ = 200–400 nm) were recorded in triplicate using the Skanlt 3.2 function on a Multiskan GO UV spectrophotometer (Thermo Scientific, Waltham, MA, USA). Distilled water was used as a blank. A minimum of three independent experiments were carried out per extract dilution. The absorbance values were recorded for wavelength intervals of 10 nm. The average absorbance values and their corresponding standard errors were plotted using the program ggplot2 of the R platform [66].

### 4.4. In Vitro Photoprotection Efficacy

We estimated UVB photoprotection efficacy using the sun protection factor (SPF_in vitro_) described by Sayre et al. [67] and by further simplification to the UV spectrophotometric Mansur´s method [68]: SPF_spectrophotometric_ = $\mathrm{CF}\;\times \sum_{290}^{320}\mathrm{EE}\left(\lambda \right)\times \;I\left(\lambda \right)\times \;A\left(\lambda \right)$, where EE(λ)—erythemal effect spectrum at wavelength λ, I(λ)—solar intensity spectrum at wavelength λ, A(λ)—absorbance of the extract solution determined by UV spectrophotometry at a wavelength (λ), and CF—correction factor (CF = 10). The values of EE (λ) × I were constant [67]. The broad-spectrum protection efficacy was determined by calculating critical wavelength [69]: λc = $\int_{290}^{\lambda c}A\left(\lambda \right)d\lambda \;=0.9\;\int_{290}^{400}A\left(\lambda \right)d\lambda$, where A(λ) was the absorbance at wavelength λ, λc is the critical wavelength (nm), and dλ is the wavelength step (1 nm). The SPF_in vitro_ values were classified in categories according to European Commission recommendations [70], as follows: no protection (0.0 ≤ SPF_in vitro_ ≤ 5.9), low protection (6.0 ≤ SPF_in vitro_ ≤ 14.9), medium protection (15.0 ≤ SPF_in vitro_ ≤ 29.9), high protection (30.0 ≤ SPF_in vitro_ ≤ 59.9), and very high protection (SPF_in vitro_ ≥ 60.0). The broad-spectrum protection sunscreens were those that showed protection efficacy values (SPF ≥ 15 and λc > 370 nm) according to the Food and Drug Administration (FDA) rule [71].

### 4.5. In Vitro Photostability

Extract aliquots (1 mL) were distributed into Petri plates with a 5 cm diameter for their irradiation. The Petri plates were irradiated in darkness using a UVA/UVB irradiation chamber BS-02 (Dr. Grobel UV-Elektronik GmbH, Etlingen, Germany) equipped with a radiation controller, UV-MAT, from the same commercial house. This radiation controller continuously measured the irradiance, calculated the irradiation dose, and switched the lamps after reaching the target dose. Operating at 100% intensity, the UVB lamps in the irradiation chamber had an irradiance value of 4 mW/cm^2^. The UVB radiation doses applied were those corresponding to minimum erythema dose (MDE) in humans according to the Fitzpatrick skin scale [24]. These were as follows: type I (0.035 J/cm^2^ = 350 J/m^2^), type II (0.056 J/cm^2^ = 560 J/m^2^), type III (0.070 J/cm^2^ = 700 J/m^2^), and type IV (0.084 J/cm^2^ = 840 J/m^2^). The relative photostability of the extracts was expressed as the percentage of effectiveness (E_ff_) of SPF_in vitro_ after UV exposure and was calculated as follows: $E_{\mathrm{ff}}=\frac{{\mathrm{SPF}}_{\mathrm{in}\;\mathrm{vitro}}\;\mathrm{after}\;\mathrm{irradiation}}{{\mathrm{SPF}}_{\mathrm{in}\;\mathrm{vitro}}\;\mathrm{before}\;\mathrm{irradiation}}\times 100$ [14].

### 4.6. Antigenotoxicity against UVB Radiation Estimates Based on SOS Chromotest

Before extract antigenotoxic effects were assayed, their genotoxicities were investigated using the SOS Chromotest [72], as was described previously by Quintero et al. [73]. The antigenotoxicity assay was conducted using a co-incubation procedure as described by Fuentes et al. [74]. Briefly, the cells were simultaneously treated with extracts (between 62.5 and 750.0 μg/mL) and 10 J/m^2^ of UVB radiation, which largely induced the SOS function *Escherichia coli* PQ37 cells [75]. After, they were cultured for 2 h at 37 °C and shaken at 300 rpm in a Thermomixer apparatus (Eppendorf, Sao Paulo, Brazil). Negative (distilled water) and positive (10 J/m^2^ of UVB) controls were always included in each assay. A minimum of four independent experiments per treatment with two replicates were conducted. β-Galactosidase (βG) and alkaline phosphatase (AP) activities were assayed in 96-well plates (Brand-GMBH, Wertheim, Germany), as described by Fuentes et al. [74]. The antigenotoxicity (the ability of the plant extract to protect against UV-induced DNA damage) was measured as a significant reduction in the SOS induction factor (*I*) in *E. coli* PQ37 cells and was expressed as a percentage of the genotoxicity inhibition, as follows: %GI = 1 − ($I_{ct}-I_{nt}$)/($I_{UVB}-I_{nt}$) × 100, where *I_ct_* was the SOS induction factor in the co-incubation procedure; *I_nt_* was the SOS induction factor in non-treated cells, and *I_UVB_* was the SOS induction factor in UVB-treated cells. Negative values of %GI were considered as zero; therefore, this parameter ranged from 0% to 100%. The minimal concentration that produces a significant (*p* ≤ 0.05) genotoxicity inhibition (CGI) in PQ37 cells was used for comparison of the genoprotective potential of the tested samples.

### 4.7. Extract Cytotoxicity in Human Fibroblast (MRC5) Cells

Cytotoxicity of flower extracts in MRC-5 cells was evaluated using the trypan blue exclusion assay [76]. The lung embryo fibroblast (MRC5) cells [77] were grown on DMEM/F-12 medium (5 mL) supplemented with 10% heat-inactivated fetal bovine serum (FBS) and 1% of GIBCO Penicillin-Streptomycin, at 37 °C, and under CO_2_ (5%) conditions in a Midi 40 incubator (Thermo Scientific, Marietta, OH, USA). Every three days, cells were grown in fresh medium to reach a confluence of 80%. Cell cultures were mixed with each extract at final concentrations between 62.5 and 750.0 μg/mL, and kept at 37 °C (24 h) under CO_2_ (5%) atmosphere conditions. After 24 h, trypsin EDTA-treated cells were centrifuged (4000× *g*, 6 min), dissolved in PBS buffer (100 μL), and mixed (10 μL) with the same volume of trypan blue (0.4%) to assess cell viability. Live and dead cells were counted using a Neubauer chamber and Eclipse E200 optical microscope (Nikon Instruments Inc., NY, USA). At least three independent experiments were carried out for each treatment. The results are expressed as percentages of cell viability (%CV) per treatment, as follows: %CV = (Living cells)/(Total cells) × 100. Lethal concentrations 50% (LC_50_) and 30% (LC_30_) for each flower extract were calculated by interpolation using the graphic method [78]. LC_50_ and LC_30_ were considered as bordering cytotoxic and non-cytotoxic concentrations, respectively. That is, extracts were cytotoxic at values ≥ LC_50_ and non-cytotoxic at values ≤ LC_30_.

### 4.8. Extract Genotoxicity in Human Fibroblast (MRC5) Cells

Genotoxicity of flower extracts in MRC-5 cells was evaluated using the Comet assay. For this purpose, a high-throughput Trevigen CometChip^®^ platform (Gaithersburg, MD, USA) was used as indicated by Sykora et al. [79], with some minor modifications. Firstly, a CometChip^®^ slide, cleaned previously with ethanol, was covered with an agarose solution prepared in PBS at 1.3% and tempered at 45 °C; then, the agarose was solidified for 24 h at 4 °C. Trypsin-treated cells (3 mL) were collected by centrifugation (2000× *g*, 6 min), were washed twice using NaCl solution (0.75%), centrifuged and suspended in fresh NaCl solution (3 mL), and then were quantified using a Neubauer counting chamber. A cell suspension (3 mL) prepared at 4.4 × 10^4^ cell/mL was mixed with an equal volume of low-melting-point agarose prepared in PBS at 1%, and the mix was poured on the CometChip^®^ slide. The agarose was solidified for 15 min at 4 °C. Finally, the CometChip^®^ chamber was ensembled by hermetically sealing to prevent mixing between the wells.

For cell treatments, 100 μL extract samples (prepared between 62.5 and 750.0 μg/mL) or standard mutagen used as a positive control (4-nitroquinoline 1-oxide prepared at 0.89 μg/mL) were loaded into wells of the CometChip^®^ chamber. A sample of DMEM medium (100 μL) was considered as the negative control. The chamber with treatments was incubated for 30 min at 37 °C under CO_2_ (5%) atmosphere conditions. The solutions of each treatment were removed from the wells, and 30 μL of Bioultra Proteinase K (0.19 mg/mL) was loaded in each well for cell enzymatic lysis (1 h, 37 °C).

After that, the CometChip^®^ slide was removed from the CometChip^®^ chamber and submerged for 15 min at 4 °C in a Comet electrophoresis tank (Cleaver Scientific Ltd., Rugby, Warwickshire, UK), which contained alkaline buffer (0.3 N NaOH, 1 mM EDTA, pH 13). The electrophoresis was carried out for 30 min at 300 mA and 25 V. The CometChip^®^ slide was submerged for 15 min in a tray containing neutralizing solution (0.4 M TRIS, pH 7.5); then, it was washed with distilled water and dried at 37 °C in a Midi 40 incubator. Finally, cell nuclei contained in each microgel were stained with 7 µL of YOYO solution (1 mM prepared in 5% DMSO) and were immediately scored using a Zeiss Axio Observer 7 fluorescence microscope (GmbH, Oberkochen, Germany).

DNA damage was expressed in arbitrary units based on the classification of Comets into five categories (0–4), as was proposed by Collins et al. [80]. A genetic damage index (GDI) was calculated for each treatment, as follows: GDI = (N_0_ × 0 + N_1_ × 1 + N_2_ × 2 + N_3_ × 3 + N_4_ × 4)/n, where Ni was the number of nuclei scored in each category and n was the number of scored cells per slide [81]. Two hundred cells per slide and two slides per treatment were analyzed. The results from at least three independent experiments were averaged to obtain the GDI values for each treatment.

### 4.9. UHPLC–ESI+–Orbitrap–MS Analysis

The flower extracts were analyzed by a UHPLC Dionex™ UltiMate™ 3000 (Thermo Fisher Scientific, Bremen, Germany) coupled to an Orbitrap™ mass detector (Exactive Plus, TFS, Bremen, Germany), using a heated-electrospray interface (HESI-II) operated in positive-ion acquisition mode (350 °C). The extract component separation was carried out on a Hypersil GOLD™ aQ column (TFS, Sunnyvale, CA, USA), of 100 mm × 2.1 mm id, ×1.9 μm particle size, at 30 °C. The mobile phase was as follows: A: water (0.2% formic acid) and B: acetonitrile (0.2% formic acid). Analysis started with 100% A and changed linearly up to 100% B in 8 min, remained for 4 min, and then returned to 100% A in 1 min; then, it remained in equilibrium for 3 min. The mobile phase flow was 0.3 mL/min, and the injection volume was 1 μL. Capillary voltage (3.5 kV, 320 °C) and higher-energy-collisional dissociation cell (HCD) were used in 10–40 eV range. Mass range in all experiments was set at *m*/*z* 80–1000. The data obtained were processed with the Thermo XCalibur™ Roadmap software, version 3.1.66.10. Compound identification was based on the extracted ion currents (EICs) of the protonated molecules, the exact masses of the ions, mass spectra fragmentation patterns, and by comparison of the experimental mass spectra with those of standards compounds.

### 4.10. Statistical Analysis

The SPF_in vitro_, λc, E_ff_, %GI, survival (%), and GDI values and their corresponding standard errors were calculated. In all cases, the data passed the Kolmogorov–Smirnov and F-maximum tests for normality and variance homogeneity, respectively; therefore, the parametric tests were used in subsequent data analyses. When a significant F-value was obtained in one-way analysis of variance (ANOVA), the groups were subsequently compared with Tukey’s test. The Pearson correlation analysis was used to examine the relationship between extract and compounds concentrations, SPF_in vitro_, and %GI estimates. For all statistical analyses, a value of *p* < 0.05 indicated significance. The R program [66] was used for all analyses.

## 5. Conclusions

Our findings show that flower extracts from ornamental plants were rich in photoprotective compounds. Several studied flower extracts showed solar filter activity and photoprotection, which was attributed to their major compounds or to their combinations. *P. latifolia* extract, which is photostable and safe, appeared to be the best candidate for use as a sunscreen active ingredient. *R. centifolia* extracts showed between low and moderate cytotoxicity and genotoxicity in human fibroblasts at the highest concentrations assayed, while the *I. horsfalliae* extract was unequivocally cytotoxic and genotoxic. Therefore, *R. centifolia* and *I. horsfalliae* extracts should be used with caution until more details on their toxicity and genotoxicity are obtained. In addition, it is necessary to test these phytochemicals using in vivo mammalian assays for practical use of these extracts in photoprotection. Sunscreen based on phytochemicals will require cost-effective processes that combine plant tissue culture and enriched-fraction extraction techniques, to guarantee a stable supply of these raw materials.

## Figures and Tables

**Figure 1 molecules-27-05525-f001:**
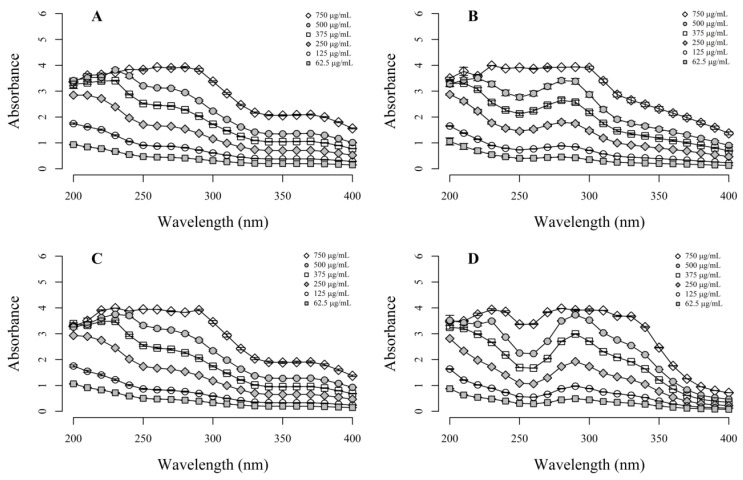
UV absorbance spectra of the flower extracts obtained from: (**A**) *R. centifolia* pink, commercial variety, (**B**) *R. centifolia* fuchsia, commercial variety, (**C**) *P. latifolia*, and (**D**) *I. horsfalliae*. Error bars indicate the standard error of the mean for at least three independent experiments (*n* = 3).

**Figure 2 molecules-27-05525-f002:**
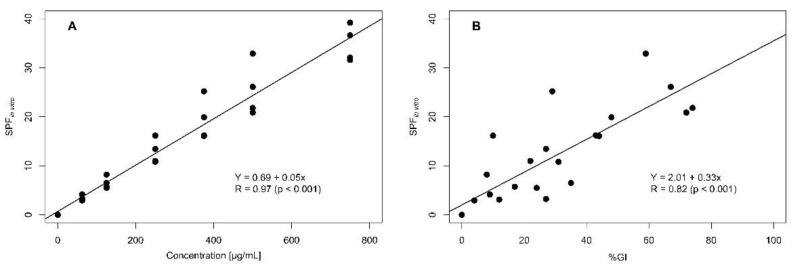
Correlation between UVB photoprotection efficacy (SPF_in vitro_) and extract concentration (**A**) and %GI estimates (**B**). A database containing 28 paired SPF_in vitro_ and %GI values, corresponding to photoprotective flower extracts, was used.

**Figure 3 molecules-27-05525-f003:**
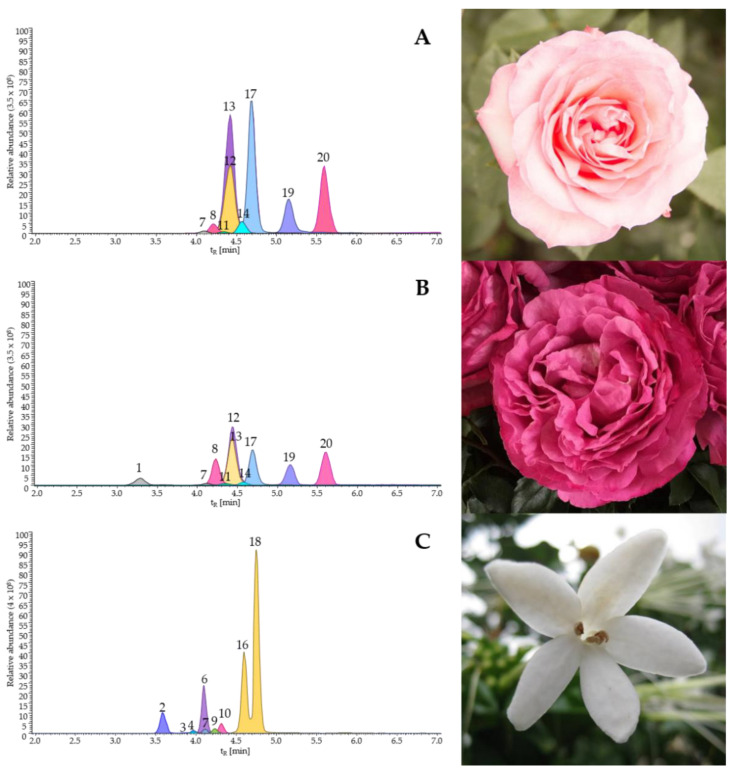
The extracted ionic currents (EICs) of [M]^+^ or protonated molecules [M + H]^+^ present in the total ion current (TIC) obtained by UHPLC–ESI+–Orbitrap–MS for three flower extracts: (**A**) *R. centifolia* (pink variety), (**B**) *R. centifolia* (fuchsia variety), and (**C**) *P. latifolia*. The peak numbers correspond to major compounds as follows: 1—Cyanidin-3,5-glucoside; 2—Chlorogenic acid; 6—Ecdysterone; 8—Quercetin-glucoside; 10—Rhamnetin-rutinoside; 12—Quercetin-3-rhamnoside; 13—Kaempferol-3-glucoside; 16—*cis*-Resveratrol-diglucoside; 17—Kaempferol-rhamnoside; 18—*trans*-Resveratrol-diglucoside; 19—Quercetin; 20—Kaempferol.

**Table 1 molecules-27-05525-t001:** List of the studied plant species. For each plant extract, the following indices are shown: (i) UVB protection efficacy (SPF_in vitro_); (ii) critical wavelength (λc); (iii) genotoxicity inhibition percentage (%GI) obtained using the SOS Chromotest; and (iv) percentages of human fibroblast cell viability (%CV). At photoprotective and non-cytotoxic extract concentrations, the percentage of effectiveness (E_ff_) values was also estimated for the minimum erythema dose (MDE) according to the Fitzpatrick skin scale.

Species (CNH Voucher)	Conc. (µg/mL)	SPF_in vitro_	λc	%GI	%CV	E_ff I_	E_ff II_	E_ff III_	E_ff IV_
*Rosa centifolia* pink, commercial variety	0	0 ± 0	0 ± 0	0 ± 0	93 ± 2	-	-	-	-
	62	3 ± 0	360 ± 0	4 ± 4	93 ± 2	-	-	-	-
	125	6 ± 0	360 ± 0	17 ± 6	91 ± 2	-	-	-	-
	250	11 ± 0	360 ± 0	31 ± 7	88 ± 2	95%	98%	100%	97%
	LC_30_ = 363	15 ± 0	360 ± 0	44 ± 3	70 ± 0	93%	100%	100%	100%
	LC_50_ = 492	21 ± 0	360 ± 0	50 ± 1	50 ± 0	83%	96%	87%	84%
	500	21 ± 0	360 ± 0	56 ± 2	49 ± 16	-	-	-	-
	750	32 ± 0	370 ± 0	72 ± 2	22 ± 10	-	-	-	-
*Rosa centifolia* fuchsia, commercial variety	0	0 ± 0	0 ± 0	0 ± 0	94 ± 1	-	-	-	-
	62	3 ± 0	357 ± 0	12 ± 8	93 ± 1	-	-	-	-
	125	5 ± 0	353 ± 0	24 ± 5	91 ± 1	-	-	-	-
	250	10 ± 0	360 ± 0	22 ± 7	87 ± 3	88%	92%	99%	100%
	LC_30_ = 492	23 ± 0	360 ± 0	30 ± 5	70 ± 0	88%	100%	84%	90%
	500	25 ± 0	360 ± 0	43 ± 3	69 ± 13	73%	84%	80%	79%
	LC_50_ = 702	36 ± 1	370 ± 0	55 ± 3	50 ± 0	91%	94%	89%	94%
	750	32 ± 1	370 ± 0	74 ± 3	45 ± 6	-	-	-	-
*Posoqueria latifolia* (COL512080)	0	0 ± 0	0 ± 0	0 ± 0	90 ± 3	-	-	-	-
	62	3 ± 0	357 ± 0	27 ± 2	90 ± 2	-	-	-	-
	125	6 ± 0	360 ± 0	27 ± 2	90 ± 1	-	-	-	-
	250	13 ± 1	360 ± 0	27 ± 2	88 ± 3	81%	79%	90%	90%
	375	19 ± 0	360 ± 0	48 ± 2	84 ± 4	86%	83%	88%	90%
	500	26 ± 2	360 ± 0	58 ± 2	82 ± 6	89%	98%	90%	100%
	750	35 ± 1	370 ± 0	67 ± 2	73 ± 6	91%	93%	92%	99%
*Ipomoea horsfalliae* (COL587134)	0	0 ± 0	0 ± 0	0 ± 0	93 ± 1	-	-	-	-
	62	4 ± 0	340 ± 0	9 ± 5	89 ± 0	-	-	-	-
	125	7 ± 0	340 ± 0	8 ± 6	76 ± 5	87%	89%	90%	91%
	LC_30_ = 250	12 ± 0	340 ± 0	10 ± 4	70 ± 9	100%	93%	90%	96%
	LC_50_ = 398	39 ± 0	350 ± 0	20 ± 6	50 ± 0	99%	100%	99%	100%
	500	39 ± 0	350 ± 0	29 ± 8	36 ± 12	-	-	-	-
	750	39 ± 0	350 ± 0	59 ± 3	16 ± 5	-	-	-	-
Commercial sunscreen	0	0 ± 0	0 ± 0	0 ± 0	94 ± 0	-	-	-	-
(Eau Thermale Avène SPF 50+) ^†,‡^	465	27 ± 0	370 ± 0	0 ± 0	84 ± 1	100%	100%	100%	100%
	930	30 ± 0	370 ± 0	4 ± 0	54 ± 6	100%	100%	100%	100%
	1870	40 ± 0	380 ± 0	14 ± 1	27 ± 5	-	-	-	-
	3750	40 ± 0	380 ± 0	32 ± 1	5 ± 3	-	-	-	-
	7500	40 ± 0	380 ± 0	42 ± 0	2 ± 0	-	-	-	-
	15,000	40 ± 0	380 ± 0	67 ± 1	0 ± 0	-	-	-	-
	30,000	40 ± 0	380 ± 0	82 ± 1	0 ± 0	-	-	-	-
Titanium dioxide ^†^	0	0 ± 0	0 ± 0	0 ± 0	88 ± 2	-	-	-	-
	50	6 ± 0	380 ± 0	0 ± 0	58 ± 1	87%	83%	75%	82%
	62	6 ± 0	390 ± 0	12 ± 1	39 ± 6	-	-	-	-
	100	11 ± 0	390 ± 0	20 ± 4	19 ± 0	-	-	-	-
	125	12 ± 0	390 ± 0	32 ± 6	0 ± 0	-	-	-	-
	250	26 ± 0	390 ± 0	57 ± 7	0 ± 0	-	-	-	-
	500	40 ± 0	380 ± 0	82 ± 3	0 ± 0	-	-	-	-
	1000	40 ± 0	380 ± 0	100 ± 2	0 ± 0	-	-	-	-
	2000	40 ± 0	380 ± 0	93 ± 4	0 ± 0	-	-	-	-

CNH: Colombian National Herbarium. The SPF_in vitro_ values were classified in categories according to the European Commission recommendation as follows: no protection (0.0 ≤ SPF_in vitro_ ≤ 5.9), low protection (6.0 ≤ SPF_in vitro_ ≤ 14.9), medium protection (15.0 ≤ SPF_in vitro_ ≤ 29.9), high protection (30.0 ≤ SPF_in vitro_ ≤ 59.9), and very high protection (SPF_in vitro_ ≥ 60.0). A λc > 370 nm defines broad-spectrum protection. The MDE values were previously indicated by Valbuena et al. [24], and these are as follows: type I (0.035 J/cm^2^ = 350 J/m^2^), type II (0.056 J/cm^2^ = 560 J/m^2^), type III (0.070 J/cm^2^ = 700 J/m^2^), and type IV (0.084 J/cm^2^ = 840 J/m^2^). ^†^, For the comparison, a widely used commercial sunscreen (Eau Thermale Avène SPF 50+) and sunscreen ingredient (titanium dioxide) were included. ^‡^, The higher sunscreen concentration (*v*/*v*) evaluated was 30 mg/mL, dissolved in distilled water.

**Table 2 molecules-27-05525-t002:** Genotoxicity of the flower extracts in MRC5 human fibroblasts cells. The genetic damage index (GDI) values and their corresponding standard errors, calculated from at least three independent experiments, are given. Pearson correlation coefficients (R) showing the relationship between GDI extract concentrations are also presented.

Conc. (µg/mL)	$\mathrm{GDI}\;(\bar{x}$ ± SE) ^†^				
	*R. centifolia* (Pink)	*R. centifolia* (Fuchsia)	*P. latifolia*	*I. horsfalliae*	SEC ^‡^
0.0	0.12 ± 0.05	0.08 ± 0.02	0.20 ± 0.05	0.21 ± 0.04	0.19 ± 0.05
46.9	0.39 ± 0.04	0.21 ± 0.09	0.48 ± 0.03	0.51 ± 0.22	0.80 ± 0.28
93.7	0.46 ± 0.04	0.27 ± 0.07	0.65 ± 0.13	0.90 ± 0.25	0.84 ± 0.28
187.5	0.98 ± 0.09	0.74 ± 0.20	0.66 ± 0.08	2.20 ± 0.40	0.83 ± 0.28
375.0	1.30 ± 0.33	1.93 ± 0.28	0.99 ± 0.27	3.31 ± 0.05	0.85 ± 0.30
750.0	2.39 ± 0.52	3.55 ± 0.04	1.56 ± 0.09	3.82 ± 0.03	0.91 ± 0.28
PC	3.91 ± 0.02	3.91 ± 0.01	3.93 ± 0.02	3.93 ± 0.02	3.90 ± 0.08
R=	0.99 (*p* ˂ 0.05)	0.99 (*p* ˂ 0.05)	0.98 (*p* ˂ 0.05)	0.92 (*p* ˂ 0.05)	0.53 (*p* = 0.28)

The standard mutagen 4-nitro-quinoline-1-oxide (0.89 µg/mL) was used as positive control (PC). ^†^, The DNA damage criteria were as follows: (i) GDI values between 0 and 1 (no DNA damage); (ii) GDI values between 1 and 2 (little DNA damage); (iii) GDI values between 2 and 3 (moderate DNA damage); and (iv) GDI values between 3 and 4 (severe DNA damage). In addition, a clear dose–response relationship (concentration–DNA damage) must exist. ^‡^, SEC: GDI values for solvent (methanol) equivalent concentrations (between 39 and 618 mM) in the extracts.

**Table 3 molecules-27-05525-t003:** Major constituents identified in flower extracts using UHPLC–ESI+–Orbitrap–MS. Extract constituents were dependent on their retention times (min) in chromatograms. The extracts were as follows: A—*Rosa centifolia* pink, commercial variety, B—*Rosa centifolia* fuchsia, commercial variety, and C—*Posoqueria latifolia*.

No.	t_R_, min	Compounds	Formula	Calculated Mass		Experim. Mass.	∆ppm	HCD, eV	Product Ions		mg/g of Extract		
											$(\bar{x}$ ± SE, *n* = 3)		
				[M]^+^	[M + H]^+^				Fragment Type	*m*/*z* (I, %)	A	B	C
1	3.30	Cyanidin-3,5-glucoside ^a,b,c^	C_27_H_31_O_16_	611.1612	-	611.15987	1.37	20	[M-C_6_H_10_O_5_]^+^ [M-2C_6_H_10_O_5_]^+^ [M-2C_6_H_10_O_5_-C_8_H_6_O_3_]^+^	449.10965 (100) 287.05522 (52) 137.02374 (21)	-	34 ± 1	-
2	3.61	Chlorogenic acid ^a,b,c^	C_16_H_18_O_9_	-	355.10235	355.10384	0.76	10	[(M + H)-H_2_O]^+^ [(M + H)-C_7_H_10_O_5_]^+^ [(M + H)-C_7_H_10_O_5_-H_2_O]^+^	337.09030 (0.3) 181.04933 (2) 163.03917 (100)	-	-	35 ± 1
3	3.83	Quercetin-rutinoside-rhamnoside ^a,b^	C_33_H_40_O_20_	-	757.21856	757.21907	0.66	0	[(M + H)-C_6_H_10_O_4_]^+^ [(M + H)-C_6_H_10_O_4_-H_2_O]^+^ [(M + H)-C_6_H_10_O_4_-C_6_H_10_O_5_]^+^ [(M + H)-C_6_H_10_O_4_-C_6_H_10_O_5_-C_6_H_10_O_4_]^+^	611.16151 (31) 593.1487 (0.1) 449.10834 (3) 303.05017 (100)	-	-	1.5 ± 0.1
4	3.96	Kaempferol-rhamninoside ^a,b^	C_33_H_40_O_19_	-	741.22365	741.22209	2.10	10	[(M + H)-C_6_H_10_O_4_]^+^ [(M + H)-2C_6_H_10_O_4_]^+^ [(M + H)-2C_6_H_10_O_4_-C_6_H_10_O_5_]^+^ [(M + H)-2C_6_H_10_O_4_-C_6_H_10_O_5_-C_8_H_6_O_2_]^+^	595.16736 (13) 449.1091 (13) 287.05585 (100) 153.01669 (0.1)	-	-	3.7 ± 0.4
5	4.00	Rhamnetin-rhamnoside ^a,b^	C_34_H_42_O_20_	-	771.23421	771.23531	1.41	0	[(M + H)-C_6_H_10_O_4_]^+^ [(M + H)-C_6_H_10_O_4_-H_2_O]^+^ [(M + H)-2C_6_H_10_O_4_]^+^ [(M + H)-2C_6_H_10_O_4_-C_6_H_10_O_5_]^+^ [(M + H)-2C_6_H_10_O_4_-C_6_H_10_O_5_-C_8_H_6_O_3_]^+^	625.17715 (34) 607.16452 (0.3) 479.11913 (17) 317.06532 (100) 167.03427 (0.1)	-	-	1.7 ± 0.1
6	4.10	Ecdysterone ^a,b^	C_27_H_44_O_7_	-	481.31598	481.31476	1.22	0	[(M + H)-H_2_O]^+^ [(M + H)-2H_2_O]^+^ [(M + H)-3H_2_O]^+^ [(M + H)-4H_2_O]^+^	463.30446 (100) 445.29547 (64) 427.28415 (16) 409.27374 (1)	-	-	64 ± 8
7	4.10	Quercetin-3-rutinoside ^a,b,c^	C_27_H_30_O_16_	-	611.1612	611.16095	0.47	10	[(M + H)-C_6_H_10_O_4_]^+^ [(M + H)-C_6_H_10_O_4_-C_6_H_10_O_5_]^+^ [(M + H)-C_6_H_10_O_4_-C_6_H_10_O_5_-C_8_H_6_O_3_]^+^	465.10226 (30) 303.04836 (100) 153.01828 (7)	1.3 ± 0.1	4.5 ± 0.1	6.9 ± 0.4
8	4.20	Quercetin-glucoside ^a,b,c^	C_21_H_20_O_12_	-	465.10275	465.10321	0.98	10	[(M + H)-C_6_H_10_O_5_]^+^ [(M + H)-C_6_H_10_O_5_-C_8_H_6_O_3_]^+^	303.04956 (100) 153.01776 (2)	6.3 ± 0.7	17 ± 1	1.3 ± 0.1
9	4.31	Kaempferol-neohesperidoside ^a,b^	C_27_H_30_O_15_	-	595.16574	595.16379	3.29	10	[(M + H)-H_2_O]^+^ [(M + H)-C_6_H_10_O_4_]^+^ [(M + H)-C_6_H_10_O_4_-C_6_H_10_O_5_]^+^ [(M + H)-C_6_H_10_O_4_-C_6_H_10_O_5_-C_8_H_6_O_2_]^+^	577.15659 (0.2) 449.10574 (33) 287.05545 (100) 153.01762 (1)	-	-	8.7 ± 0.5
10	4.31	Rhamnetin-rutinoside ^a,b^	C_28_H_32_O_16_	-	625.17631	625.17697	1.06	10	[(M + H)-C_6_H_10_O_4_]^+^ [(M + H)-C_6_H_10_O_4_-C_6_H_10_O_5_]^+^ [(M + H)-C_6_H_10_O_4_-C_6_H_10_O_5_-C_8_H_6_O_3_]^+^	479.11914 (27) 317.06548 (100) 167.03214 (0.01)	-	-	17 ± 1
11	4.33	Quercetin-arabinoside ^a,b^	C_20_H_19_O_11_	-	435.09218	435.0923	0.11	10	[(M + H)-C_5_H_8_O_4_]^+^ [(M + H)-C_5_H_8_O_4_-C_8_H_6_O_3_]^+^	303.04884 (100) 153.01845 (6)	1.08 ± 0.04	1.4 ± 0.2	-
12	4.43	Quercetin-3-rhamnoside ^a,b,c^	C_21_H_20_O_11_	-	449.10838	449.10800	0.36	10	[(M + H)-C_6_H_10_O_4_]^+^ [(M + H)-C_6_H_10_O_4_-C_8_H_6_O_3_]^+^	303.04836 (100) 153.01828 (4)	49 ± 2	32 ± 1	-
13	4.43	Kaempferol-3-glucoside ^a,b,c^	C_21_H_20_O_11_	-	449.10783	449.10773	0.24	10	[(M + H)-C_6_H_10_O_5_]^+^ [(M + H)-C_6_H_10_O_5_-C_8_H_6_O_2_]^+^	287.05415 (100) 153.01802 (1)	70 ± 12	41 ± 1	-
14	4.57	Kaempferol-arabinoside ^a,b^	C_20_H_18_O_10_	-	419.09727	419.09756	0.69	0	[(M + H)-C_5_H_8_O_4_]^+^ [(M + H)-C_5_H_8_O_4_-C_8_H_6_O_2_]^+^	287.05553 (85) 153.01845 (1)	6.4 ± 0.5	2.0 ± 0.1	
15	4.60	Rosmarinic acid ^a,b,c^	C_18_H_16_O_8_	-	361.09179	361.09157	0.62	10	[(M + H)-H_2_O]^+^ [(M + H)-C_9_H_8_O_4_]^+^ [(M + H)-C_9_H_8_O_4_-H_2_O]^+^	343.07965 (0.2) 181.04958 (10) 163.03841 (100)	-	-	0.5 ± 0.0
16	4.60	*cis*-Resveratrol-diglucoside ^a,b^	C_26_H_32_O_13_	-	553.19156	553.19199	0.41	10	[(M + H)-H_2_O]^+^ [(M + H)-2H_2_O]^+^ [(M + H)-C_6_H_10_O_5_]^+^ [(M + H)-C_6_H_10_O_5_-H_2_O]^+^ [(M + H)-2C_6_H_10_O_5_]^+^	535.18141 (3) 517.17064 (0.4) 391.13762 (94) 373.12898 (100) 229.08403 (0.1)	-	-	140 ± 7
17	4.70	Kaempferol-rhamnoside ^a,b^	C_21_H_21_O_10_	-	433.11292	433.11453	1.39	10	[(M + H)-C_6_H_10_O_4_]^+^ [(M + H)-C_6_H_10_O_4_-C_8_H_6_O_2_]^+^	287.05499 (100) 153.01749 (0.3)	64 ± 8	23 ± 2	-
18	4.75	*trans*-Resveratrol-diglucoside ^a,b^	C_26_H_32_O_13_	-	553.19156	553.19199	0.41	10	[(M + H)-H_2_O]^+^ [(M + H)-2H_2_O]^+^ [(M + H)-C_6_H_10_O_5_]^+^ [(M + H)-C_6_H_10_O_5_-H_2_O]^+^ [(M + H)-2C_6_H_10_O_5_]^+^	535.18141 (8) 517.17064 (0.2) 391.13762 (100) 373.12898 (62) 229.08403 (0.1)	-	-	280 ± 16
19	5.16	Quercetin ^a,b,c^	C_15_H_10_O_7_	-	303.05047	303.0499	0.10	10	[(M + H)-C_8_H_6_O_3_]^+^	153.01776 (1)	160 ± 26	130 ± 14	-
20	5.63	Kaempferol ^a,b,c^	C_15_H_10_O_6_	-	287.05501	287.05647	5.08	10	[(M + H)-C_8_H_6_O_2_]^+^	153.01897 (0.2)	146 ± 5	51 ± 4	-

^a^ Tentative identification based on comparison with [M^+^] or [M + H]^+^ ions reported in the literature for *Rosa* spp. [25,26]. ^b^ Tentative identification based on comparison with molecule fragmentation pattern in mass spectra and on databases [27,28,29,30]. ^c^ Standard compounds used for the comparison of their mass spectra with those present in the extracts studied. HCD, higher-energy-collisional-dissociation cell.

## Data Availability

All data, tables, and figures are original.

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
