# Peer review of "Flower Extracts from Ornamental Plants as Sources of Sunscreen Ingredients: Determination by In Vitro Methods of Photoprotective Efficacy, Antigenotoxicity and Safety"

_molecules, 2022, doi:10.3390/molecules27175525_

Round 1

Reviewer 1 Report

The manuscript " Flower Extracts from Ornamental Plants as Sources of Sunscreen Ingredients: Determination by In Vitro Methods of Photoprotective Efficacy, Antigenotoxicity and Safety" is devoted to comprehensive study of the properties of Rosa x centifolia, Posoqueria latifolia, and Ipomoea horsfalliae extracts. UV-absorption capability, photoprotection efficacy, cytotoxicity and genotoxicity in human fibroblast (MRC5) cells were estimated, and UHPLC-ESI+ -Orbitrap-MS analysis was performed to obtain the chemical composition of flower extracts. The results of this work these data are important for the industry and sufficiently novelty.

The manuscript is written clearly, well-structured and has a good scientific soundness.

I think, this manuscript can be published in the Molecules journal after minor revision taking into account and some of the remarks described below:

1.      The authors compare properties of the flower extracts with commercially available sunscreen and titanium dioxide. In this case, it would be better to present more discussion of properties these reference substances. Also, it would be better to discuss time dependence of photoprotection: how long does the exposure time take for the extracts to degrade?

2.      Introduction: please, provide more information about known chemical composition of plant used.

Author Response

  1. The authors compare properties of the flower extracts with commercially available sunscreen and titanium dioxide. In this case, it would be better to present more discussion of properties these reference substances. Answer: We included a new paragraph in Discusion section on this topic.
  2. Also, it would be better to discuss time dependence of photoprotection: how long does the exposure time take for the extracts to degrade?. Answer: It is important to clarify, that the photostability of a sunscreen will depend on the UV irradiance or UV dose, and other environmental factors. All of them, must be considered to measure the final sunscreen photostability. That said, we only used minimal erythema dose (MED) to measure the relative photostability of the extracts. The MED refers to that UV dose required to produce erythema in different skin types (I - IV) according to the Fitzpatrick scale. Operating at 100% intensity, the UVB lamps in the irradiation chamber had an irradiance value of 4mW/cm2 (see Materials and Methods section); therefore, irradiation times were between 21 and 47 seconds for MED types I and IV. Under such as conditions, the photoprotective extracts resulted photostable. We think that a more realistic photostability estimate of the sunscreens should be studied in the final cosmetic formulation phase using increasing MEDs.
  3. Introduction: please, provide more information about known chemical composition of plant used.  Answer: According our knowlegment, there not previous works on chemical composition of P. latofolia and R. centifolia extracts. If reviewers know new works about this mater, please, send the links. 

Reviewer 2 Report

The manuscript by Fuentes et al. describes the evaluation of four ornamental flower extracts regarding their photoprotective potential.

It is a well written paper, with sufficient background, and the experimental design adequate to answer the research questions. The conclusions are supported by the results and I believe this is an interesting paper for the readers of Molecules.

Regarding the results, it is not clear which were the criteria used to choose which concentration of each extract/positive control was chosen to determine the percentage of effectiveness (Eff). Please clarify.

I also think that is a little strange that the % of cell viability at a concentration of 0 μg/mL is not 100%. Aren't the values obtained for the other concentrations by comparision with the negative control (i.e. 0 μg/mL). I also believe that the MTT assay would be most suitable for this evaluation than the trypan blue exclusion assay, since the first relies on the spectrophotometric reading of the formazan crystals formed, and the later is dependent on the personal counting of cells. However, it is also a valid method, so I respect the author's choice of methodology.

In line 240, please correct "specie" to species. Line 242 correct "knowns" to known.

Author Response

  1. Regarding the results, it is not clear which were the criteria used to choose which concentration of each extract/positive control was chosen to determine the percentage of effectiveness (Eff). Please clarify. Answer: The criteria was to study effectivenes only at photoprotective and safe extract concentrations. This was clarified in Results section.
  2. I also think that is a little strange that the % of cell viability at a concentration of 0 μg/mL is not 100%. Aren't the values obtained for the other concentrations by comparision with the negative control (i.e. 0 μg/mL). Answer: In theory, the negative control of an experiment (i.e. 0 μg/mL) using trypan blue should produce 100% unstained (live) cells; but in practice this does not occur even observing the sample under the microscope immediately after staining. We assumed that the error varies as a direct consequence of the researcher's delay in observing the sample after staining. In other words, it is an intrinsic method error. We clarify that in our study, all observations were made by the same researcher.
  3. I also believe that the MTT assay would be most suitable for this evaluation than the trypan blue exclusion assay, since the first relies on the spectrophotometric reading of the formazan crystals formed, and the latter is dependent on the personal counting of cells. However, it is also a valid method, so I respect the author's choice of methodology.  Answer: I agree with the reviewer that the MTT method is more widely used to measure lethality. However, lethality measurement using MTT assay can be affected when studying pigmented samples; as it is the case of the plant extracts.  In addition, we were interested in knowing the effect of the extracts on the structural integrity of the cell membrane frequently related to cell death.  For this purpose, trypan blue assay is more adequate.